# The Influence of the South-to-North Water-Diversion Project on Terrestrial Water-Storage Changes in Hebei Province

**Tianxu Liu [1], Dasheng Zhang [2], Yanfeng Shi [3], Yi Li [4], Jianchong Sun [5] and Xiuping Zhang [6],***

[1] Hydrology and Water Resources Survey Bureau of Shijiazhuang City, Shijiazhuang 050051, China; hbszy9909@163.com
[2] Hebei Provincial Institute of Water Science, Shijiazhuang 050051, China; 18501317443@163.com
[3] School of Municipal and Environmental Engineering, Shandong Jianzhu University, Jinan 250101, China; 13953@sdjzu.edu.cn
[4] School of Civil Engineering, Architecture and Environment, Hubei University of Technology, Wuhan 430068, China; liyi_bnuphd@mail.bnu.edu.cn
[5] College of Water Science, Beijing Normal University, Beijing 100875, China; jcsun2021@mail.bnu.edu.cn
[6] Jiangxi Academy of Water Science and Engineering, Nanchang 330029, China
* Correspondence: zhangxiuping81@163.com

**Abstract:** The lack of water resources has emerged as a major factor limiting the high-quality economic and ecological development in Hebei Province. Therefore, it is of great significance to understand the dynamic changes in terrestrial water storage for effectively managing water resources in Hebei Province. The evolution pattern and spatial distribution of TWS anomalies (TWSA) were analyzed utilizing gravity recovery and climate experiment (GRACE) solutions and the water balance method from 2003 to 2020, and the missing monthly data during GRACE and GRACE-FO missions were filled by combining the climate-driven model and meteorological products. Moreover, the impact of the south-to-north water-diversion (SNWD) project on alleviating the water-storage deficit was quantified. The results revealed that the water-balance method on the strength of the combination of CMA precipitation and Noahv2.1-simulated evapotranspiration and runoff data matches well with the TWSA data derived from GRACE, with a correlation coefficient up to 0.95. However, the accuracy was unsatisfactory during the process of characterizing the spatial characteristics of TWSA. After the SNWD project, GRACE-derived results showed that the downtrends of TWSA were reduced by 10.93%, especially in mountainous areas: by 79.78%. Concerning the spatial scale, the deficit trends were reduced to a certain extent in northern Hebei Province, while the decreasing trends cannot be reversed for a short time in southern areas where human activities are intensive.

**Keywords:** GRACE satellites; water-balance method; terrestrial water-storage anomalies; Hebei Province; south-to-north water-diversion project

## 1. Introduction

Terrestrial water storage (TWS) is a considerable component of the Earth's water cycle, and it refers to the total sum of water in all forms present on or beneath the surface [1,2], reflecting the net effect between inputs (such as precipitation, snow or glacier melt, and runoff) and outputs (such as human water consumption and runoff) [3]. Understanding the fluctuations in water storage is crucial for effective water resource management and the global water cycle [4,5]. However, due to the absence of in situ measurements and the limitations of traditional monitoring methods, it can often be challenging to gather continuous spatiotemporal data on water storage in practice.

Many researchers have employed the water-balance (WB) method to estimate the terrestrial water-storage anomalies (TWSA) over large areas by combining the precipitation, evapotranspiration (ET), and runoff products. For instance, Nie et al. [6] used the Global Land Data Assimilation System (GLDAS) to reconstruct the TWSA for estimating its

characteristics in the Amazon River Basin from 1948 to 2013. On account of the WB method and multi-source remote sensing data, Yin et al. [7] reconstructed the TWSA series from 1980 to 2015 in the Beishan area, Gansu Province. The findings suggested that the WB-based TWSA estimates indicated a comparable downward trend compared to gravity recovery and climate experiment (GRACE) solutions, with the rate of −0.94 mm/year. Zhang et al. [5] evaluated the spatiotemporal features of TWSA using the WB method across 10 river basins in China and validated them against GRACE solutions and GLDAS simulations during 2003–2020. The study revealed that comparable estimates of TWSA could be independently computed using the WB method in 4 out of 10 river basins. Launched jointly by NASA and DLR in March 2002, the GRACE satellite can effectively monitor the time-varying global gravity field, providing a new means to quantify changes in terrestrial water storage on regional to global scales [8,9].

Thus far, GRACE solutions have been widely used for water-storage evaluation in many typical regions of the world and achieved remarkable performance therein, such as the Amazon River Basin in the United States [10,11], India [12,13], and the North China Plain in China [14,15]. Notably, Feng et al. [16] successfully estimated the groundwater depletion in North China by combining GRACE-derived TWSA estimates and simulated soil moisture from 2003 to 2010. This research provided valuable contributions of water resources managements. However, the discontinuous TWSA time series restricts its wide applications because of the data gaps between GRACE and GRACE follow-on (GRACE-FO) missions. Many attempts have been adopted to overcome this limitation. For instance, Mo et al. [17] developed an innovative Bayesian convolutional neural network for reconstructing the missing signals between 2017–2018 from hydroclimatic inputs. Results indicated that the reconstructed signals showed R > 0.70 with the testing GRACE(-FO) solutions, accounting for more than 90% of the globe. Yang et al. [18] selected Australia as the study area and reconstructed the TWSA solutions by combining GRACE-derived TWSA trends with the detrended TWSA from April 2002 to July 2019. The reconstructed TWSA showed great consistency with GRACE solutions, with a correlation coefficient of 0.98 and Nash–Sutcliffe efficiency of 0.96. Rateb et al. [19] modeled GRACE data to fill the missing data and quantify uncertainties in the existing and missing observations based on Bayesian framework. The results indicated that the reconstructed solutions account for 99% of the variability of the original data at the basin scale and 78% at the one-degree grid scale. The above research provides insightful suggestions and datasets to achieve continuous GRACE solutions for long-term hydrological researches.

Hebei Province, being an important grain-production base in China, has suffered from prolonged and severe water shortage, which has become a major bottleneck restricting high-quality economic and social development. In addition, overexploitation of water resources has led to various environmental and geological problems, including water-quality deterioration and land subsidence [20]. To mitigate the severe water shortage in North China, China has undertaken the south-to-north water-diversion (SNWD) project, and the central route worked in December 2014. Up to December 2022, the Middle Route of the SNWD project has diverted 58.6 billion $m^3$ of freshwater resources totally from the Danjiangkou Reservoir in the Hanjiang River to the North China Plain, which has significantly improved the situation of the water resources of Hebei Province [21]. However, whether and to what extent this project has eased the decreasing trend of water storage is currently a hot and difficult issue with respect to the province.

In order to effectively solve these above problems, this study utilizes GRACE solutions and WB-based estimates to analyze the spatiotemporal characteristics of TWSA from January 2003 to December 2020 and quantitatively evaluate the influence of the SNWD central route in alleviating the loss of TWS in Hebei Province. This study can provide reliable data support and technical support for scientifically mastering the spatiotemporal variation law of water resources in Hebei Province.

## 2. Materials and Methods

### 2.1. Study Area

Hebei Province was chosen as the study area, which is distributed in the North China Plain. It is between 113°27′–119°50′ E longitude and 36°05′–42°40′ N latitude, covering an area of approximately $1.88 \times 10^5$ km$^2$, as shown in Figure 1. It has higher terrain in the northwest and lower terrain in the southeast, including plateaus, mountains, hills, and plains. Precipitation is concentrated in summer, with the multi-year mean annual precipitation of 503.49 mm [22]. Based on the Hebei Water Resources Bulletin, the water resources of the province have been deficient for many years, of which groundwater storage loss is the most serious. According to data statistics from 2010 to 2017, the average annual total water resources in Hebei Province is $161.79 \times 10^8$ m$^3$, while the average water consumption is $190.05 \times 10^8$ m$^3$, indicating a shortage in the overall water supply. The average per capita water resource is 220.12 m$^3$, which is only 10.51% of the national per capita level [20]. The plains have seen several groundwater funnels, including two shallow groundwater funnels and two deep groundwater funnels, such as Jizaoheng and Nangong (http://slt.hebei.gov.cn, accessed on 15 January 2020). The south-to-north water-diversion project was implemented in 2002 for relieving the shortage of water resources in North China, including three routes (east, central, and west), of which the central route officially diverted water to North China (including Beijing, Tianjin, Hebei, and Henan) in December 2014, with 9.21 billion km$^3$ of water transported from Danjiangkou Reservoir to northern China each year.

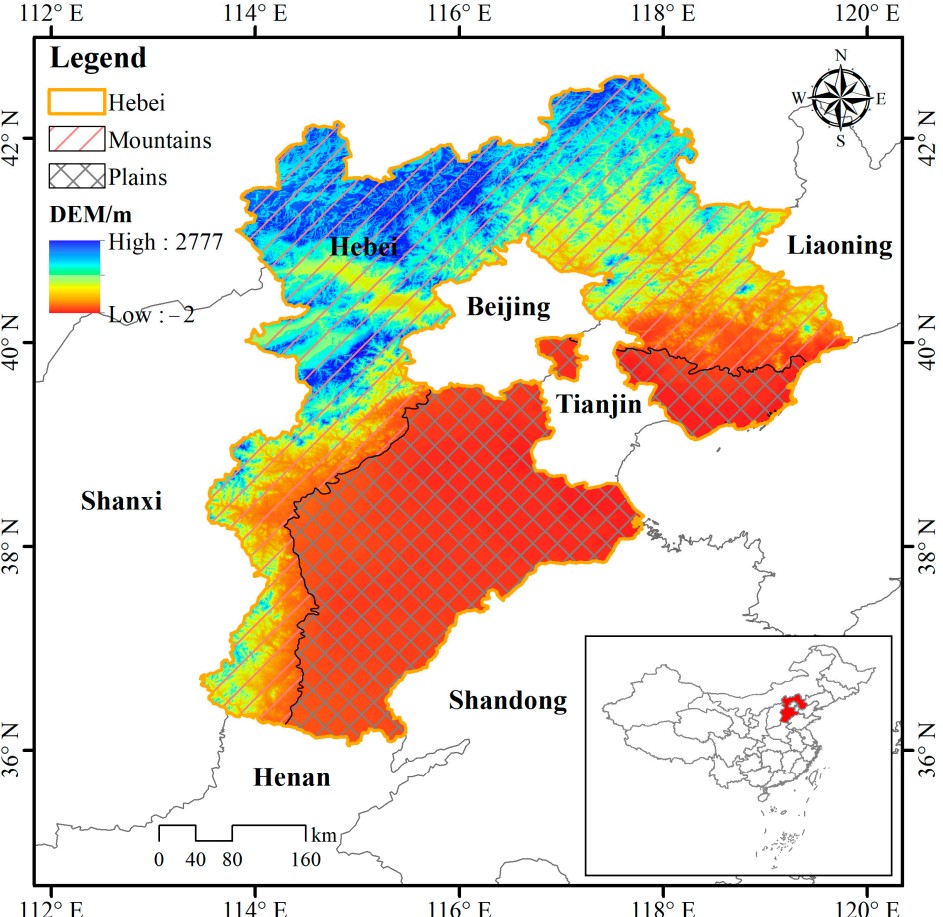

**Figure 1.** Surface elevation and geographical location in the study area.

*2.2. Datasets*

2.2.1. GRACE Datasets

The CSR Mascon RL06 (CSR-M06) and JPL Mascon RL06 (JPL-M06) grid products are released by the Center for Space Research (CSR) at the University of Texas at Austin and the Jet Propulsion Laboratory (JPL) of NASA, respectively. The study period spans from January 2003 to December 2020. The spatial resolution of CSR solutions is $0.25° \times 0.25°$ and $0.5° \times 0.5°$ for JPL. The products represent TWSA relative to the mean baseline between January 2004 and December 2009. The JPL-M06 product uses officially published scale factors to correct the leakage errors, while the CSR-M06 product does not require additional processing [23,24]. A few monthly data were missing in the later period in order to extend the operating life of the GRACE satellite. The data gaps of GRACE/GRACE-FO were filled based on the cubic spline interpolation [25], whereas the one-year gap between the two missions was reconstructed employing the method raised in this study.

2.2.2. Meteorological Data

Precipitation data are offered by the National Meteorological Science Data Center (http://data.cma.cn/, CMA) [26] and the Global Precipitation Measurement (GPM) [27]. Evapotranspiration (ET) products were obtained from the Global Land Evaporation Amsterdam Model (GLEAM) [28,29]. In addition, the precipitation, evapotranspiration, and runoff data required for this study were provided by the Noahv2.1 model of GLDAS [30]. Due to the discrepancies of the spatial resolution, all products were resampled to the same spatial resolution as GRACE observations so as to facilitate the calculation and analysis of the spatial characteristics of water storage. A detailed introduction of the above data is shown in Table 1.

**Table 1.** Information of multi-source products used in the research.

| Dataset | Sources | Spatial Resolution | Website |
|---|---|---|---|
| TWSA | CSR-M06<br>JPL-M06 | $0.25° \times 0.25°$<br>$0.50° \times 0.50°$ | http://www2.csr.utexas.edu/<br>https://podaac-opendap.jpl.nasa.gov/ |
| Precipitation | GPM<br>CMA<br>Noahv2.1 | $0.10° \times 0.10°$<br>$0.50° \times 0.50°$<br>$0.25° \times 0.25°$ | https://disc.gsfc.nasa.gov/<br>http://data.cma.cn/<br>https://disc.gsfc.nasa.gov/datasets |
| ET | GLEAM<br>Noahv2.1 | $0.25° \times 0.25°$<br>$0.25° \times 0.25°$ | https://www.gleam.eu<br>https://disc.gsfc.nasa.gov/datasets |
| Runoff | Noahv2.1 | $0.25° \times 0.25°$ | https://disc.gsfc.nasa.gov/datasets |

2.2.3. Auxiliary Data

The temperature and precipitation products are provided by CN05.1 so as to reconstruct the climate-driven TWSA estimates. The temporal and spatial resolution are daily and 0.25°, respectively. The CN05.1 products are developed based on the "anomaly approach" in the process of interpolation [31]. Previous studies [32] have demonstrated that the input variables based on CN05.1 can generate more accurate TWSA solutions than other remote sensing products and were thus used in this study. Up to now, many innovative methods have been developed to reconstruct long-term TWSA solutions, including long-term precipitation driven [33], Bayesian convolutional neural network [34], and Bayesian framework (BF for short) [35]. The BF-based TWSA solutions have been proven to display the best performance in the Haihe River Basin relative to other available reconstructed results; thus, this product was utilized to assess the accuracy of the reconstructed results computed by this research.

### 2.3. Methods for Estimating TWSA

#### 2.3.1. Water-Balance (WB) Method

The WB method is a balance calculation of the quantitative relationship between water supply and drainage in a region over a specific period of time. Based on the theory of water balance, the instantaneous change in terrestrial water storage (*TWSC*) is equal to the difference between the recharge and discharge in the region, where the recharge source is mainly rainfall, and the discharge mainly contains runoff (*R*) and *ET*, as depicted in formula (1) [36].

$$TWSC = P - R - ET \tag{1}$$

*TWSC* can also be obtained based on the water-balance method through Formulas (2) and (3) [37]:

$$TWSC = \frac{TWSA(t) - TWSA(t-1)}{\Delta t} \tag{2}$$

$$TWSA(t) = TWSC \cdot \Delta t + TWSA(t-1) \tag{3}$$

where *P* stands for the monthly precipitation (mm/month); *R* means the runoff (mm/month); *ET* represents evapotranspiration (mm/month); $\Delta t$ denotes the time interval, which is usually one month.

#### 2.3.2. Climate-Driven TWSA Reconstruction

From the perspective of TWSA changes, the TWSA retrieved by GRACE is affected by the comprehensive influence of meteorological change and human activities [28], while the joint changes in precipitation and temperature, as basic natural variables, can be an approximate representation of global or regional TWSA changes under natural conditions [38]. Inspired by the fundamental principles of hydrological simulation, Humphrey and Gudmundsson built a climate-driven day-scale TWSA through establishing a statistical model with precipitation and temperature as the driving variables, and the signal connection between climate-driven day-scale TWSA and GRACE TWSA is used for scaling transformation processing to achieve the purpose of rebuilding the total day-scale TWSA with GRACE TWSA as the constraint data [21], as follows in the given formulas:

$$TWSA(t) = (TWSA(t-1)) \cdot e^{\frac{-1}{\tau(t)}} + P(t) \tag{4}$$

$$\tau(t) = a + b \cdot T_z(t) \tag{5}$$

$$T_z = 1 - \tanh\left(\frac{T_0 - mean(T_0)}{sd(T_0)}\right) \tag{6}$$

$$T_0 = \begin{cases} 0, & T < 0 \\ T, & T \geq 0 \end{cases} \tag{7}$$

where *P*(*t*) and *t* represent the daily precipitation input and the daily time vector relative to 2004, respectively, and $e^{\frac{-1}{\tau(t)}}$ is the attenuation function of water storage, which is temperature-dependent and takes the value (0, 1). *TWSA*(0) takes the average of GRACE TWSA in 2004. In order to assure the accuracy of the reconstructed day-scale results, the results driven by precipitation and temperature were averaged into months, then de-trended and de-cycled, and were processed. Finally, GRACE TWSA, which was also de-trended and de-cycled, was used for constraints [21]:

$$anom(GRACE(t_m)) = \beta \cdot anom(TWSA(t_m)) + \varepsilon \tag{8}$$

where $\beta$ is the constraint factor (also called the correction parameter), $\varepsilon$ represents the error, $t_m$ denotes the monthly time vector relative to 2004, and *anom* denotes the subtraction of

the trend and period terms from the original series. Among them, the parameters *a*, *b*, and *β* involved in formulas were obtained by the Markov chain Monte Carlo (MCMC) method after tens of thousands of simulations [21].

### 2.3.3. Mann–Kendall (MK) Trend Test

The MK test is a nonparametric statistical test method that was proposed by Mann (H.B. Mann) and Kendall (M.G. Kendall) [39]. The data do not have to follow a normal distribution and are not subject to the influence of outliers in this method. Thus, the MK test has been widely utilized to detect trends in both hydrological and climatic series [40].

Assume, H0, that the data in the sample are random variables and satisfy independent and identical distributions with no linear trend. The test statistic is given:

$$S = \sum_{i=1}^{n-1} \sum_{j=+1}^{n} sgn(x_j - x_i) \tag{9}$$

where *n* denotes the sum of samples, and $x_i$ and $x_j$ stand for the *i*th and *j*th values of the time series, respectively. The symbolic function $sgn(x_j - x_i)$ is defined as follows:

$$sgn(x_j - x_i) = \begin{cases} +1 & if(x_j - x_i) > 0 \\ 0, & if(x_j - x_i) = 0 \\ -1, & if(x_j - x_i) < 0 \end{cases} \tag{10}$$

When *n* < 10, the test statistic *S* can be directly tested for two-sided trends; when $n \leq 10$, the test statistic *S* follows the standard normal distribution, and the test statistic is constructed as given:

$$Z = \begin{cases} \frac{S-1}{\sqrt{Var(S)}}, & S > 0 \\ 0, & S = 0 \\ \frac{S+1}{\sqrt{Var(S)}}, & S < 0 \end{cases} \tag{11}$$

where *Var* (*S*) denotes the variance of *S*, which is related to whether there are duplicates of the variables in the sample.

$$Var(S) = \begin{cases} \frac{n(n-1)(2n+5)}{18}, & p = 0 \\ \frac{\left[n(n-1)(2n+5) - \sum_{p=1}^{g} t_p(t_p-1)(2t_p+5)\right]}{18}, & p \neq 0 \end{cases} \tag{12}$$

where *p* is the quantity of duplicates, *g* is the quantity of unique values, and *tp* represents the quantity of repetitions of each duplicate. Under the condition that the significance level *α* is given, when $Z \geq |Z_{1-\alpha/2}|$, we reject the original hypothesis *H*0, indicating that there is a remarkable tendency in the sequence. This study utilizes the significance levels *α* = 0.01 and 0.05. If *Z* > 1.96, it reveals significance at the 0.05 level. If *Z* > 2.576, it indicates significance at the 0.01 level.

### 2.3.4. Evaluation Indexes

In this paper, we mainly use correlation coefficient (CC), Nash–Sutcliffe efficiency (NSE), root mean square error (RMSE), and mean absolute error (MAE) to evaluate the quality of model reconstruction results, and the four metrics are calculated as follows [41,42]:

$$CC = \frac{\sum_{i=1}^{n}(X_i - \overline{X})(Y_i - \overline{Y})}{\sqrt{\sum_{i=1}^{n}(X_i - \overline{X})^2}\sqrt{\sum_{i=1}^{n}(Y_i - \overline{Y})^2}} \tag{13}$$

$$NSE = 1 - \frac{\sum_{i=1}^{n}(Y_i - X_i)^2}{\sum_{i=1}^{n}(X_i - \overline{X})^2} \tag{14}$$

$$RMSE = \sqrt{\frac{1}{n}\sum_{i=1}^{n}(Y_i - X_i)^2} \qquad (15)$$

$$MAE = \frac{1}{n}\sum_{i=1}^{n}|Y_i - X_i| \qquad (16)$$

where $Y$ stands for the measured result; $X$ denotes modelled values; $\overline{Y}$ and $\overline{X}$ mean the average of the observed and modelled values, respectively; and $n$ denotes the total number of data. The reconstructed results are more accurate the greater the *NSE* and *CC* are between the observed and simulated values. The accuracy of the model increases as the *RMSE* and *MAE* approach zero.

2.3.5. Research Framework

The flowchart is depicted as shown in Figure 2, and detailed process and descriptions are as follows:

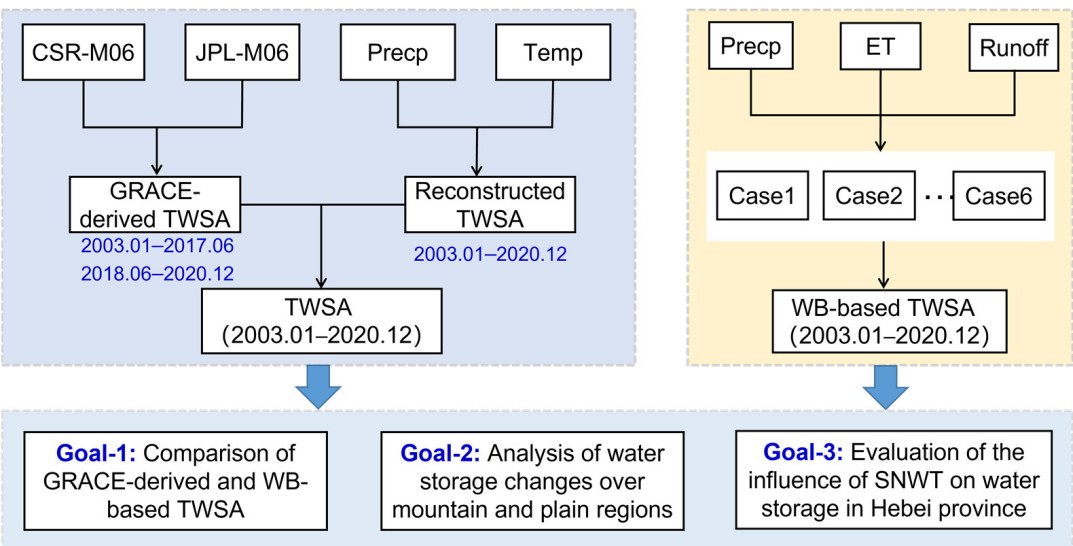

**Figure 2.** Flowchart of this study.

Firstly, two kinds of GRACE mascon solutions were utilized to calculate TWSA series during the period from 2003 to 2020, which include CSR-M06 and JPL-M06, respectively. Moreover, the uncertainties of two mascon solutions were quantified. In order to bridge the missing data between GRACE/GRACE-FO missions, the TWSA were reconstructed by combining precipitation, temperature, and the climate-driven reconstructed methods, as depicted in Section 2.3.2. Thus, the full-time TWSA series was generated in the regional and pixel scales.

Secondly, different kinds of meteorological variables were applied to calculate the monthly TWSA solutions based on the WB method, provided by GPM, CMA, GLEAM, and Noahv2.1. A total of six cases was generated based on the meteorological data. Then, the WB-based TWSA was provided to characterize the spatiotemporal changes in water storage over the Hebei Province between 2003–2020.

Lastly, based on these above results, the accuracy of WB-based TWSA was validated based on GRACE-derived results from the viewpoint of temporal and spatial scales. The characteristics of TWSA were analyzed by using GRACE- and WB-based results. More importantly, this study further evaluates the influence of the SNWD project on the water storage over mountain regions, plain regions, and Hebei Province to provide suggestions and data support for the management of water resources.

## 3. Results

### 3.1. Evaluation of Different Meteorological Products

The time series of three precipitation and two ET products are shown in Figure 3, spanning from January 2003 to December 2020 over Hebei Province. Figure 3a shows that the seasonal characteristics and variation ranges of precipitation data are basically consistent, with the CC and NSE above 0.96 (Figure 4), and 74% of precipitation is concentrated from June to September of each year. The differences between GPM and CMA precipitation data are the smallest, and the RMSE and MAE metrics are 6.19 mm and 3.18 mm, respectively. The differences between Noahv2.1 and CMA precipitation data are the largest, with the RMSE and MAE values of 9.41 mm and 6.02 mm (Figure 4). With respect to the ET products, it can be observed from Figure 3b that the seasonal characteristics of the two ET data show good consistency, with CC and NSE of 0.96 and 0.71, respectively. In addition, the variation range of the Noahv2.1 ET product is significantly larger than that of the GLEAM result. The maximum differences mainly occur in summer. Specifically, the RMSE and MAE between the two ET products are 14.06 mm and 10.70 mm, respectively.

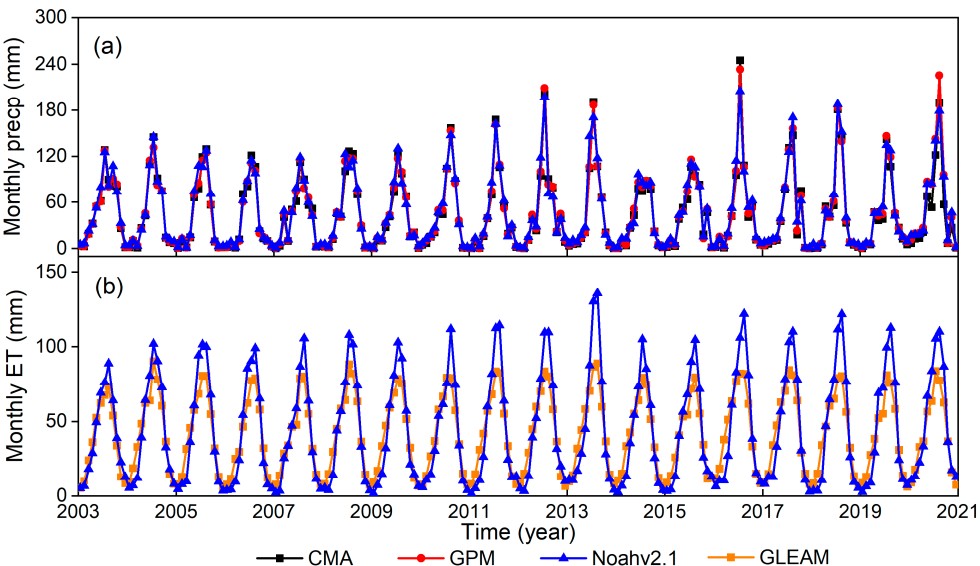

**Figure 3.** Comparisons of different kinds of (**a**) precipitation and (**b**) ET products over Hebei Province from 2003–2020.

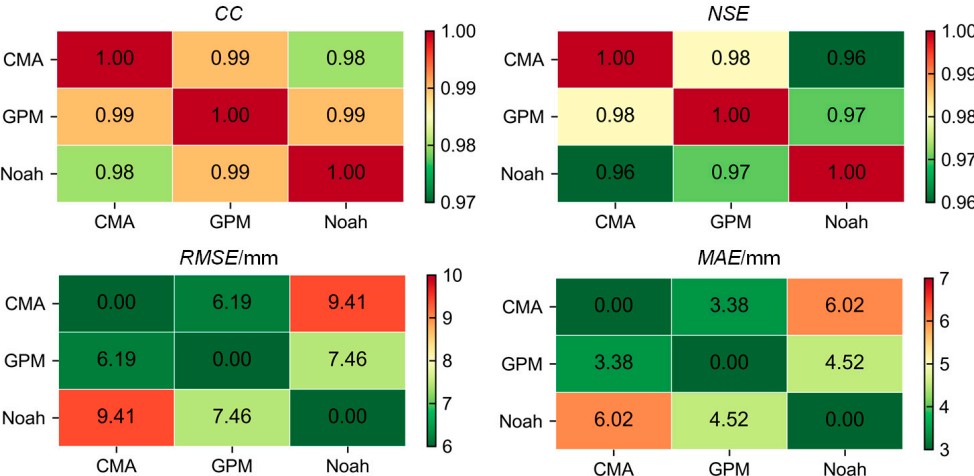

**Figure 4.** Heatmap of all kinds of precipitation data in Hebei Province between 2003 and 2020.

### 3.2. Reconstruction of TWSA in Hebei Province

Figure 5a displays the results of TWSA in Hebei Province from 2003–2020 as determined by different GRACE mascon solutions. The shaded areas stand for the uncertainties of mascon solutions. It can be observed that CSR-M06 and JPL-M06 solutions show significant downward trends with the slope of $-13.70 \pm 0.51$ mm/year and $-16.44 \pm 0.41$ mm/year, individually. This indicates that the TWSA in Hebei Province are in a state of continuous loss during 2003–2020. Moreover, these two solutions exhibit strong agreement with a high correlation coefficient of 0.96. The shaded areas reflect the uncertainty of the two results. The uncertainty of CSR-M06 result is 2 cm, as stated on the official website (http://www2.csr.utexas.edu/grace/RL06.html, accessed on 15 January 2020), while the uncertainty of JPL-M06 result was calculated based on the data provided by the official website. To minimize the influence of data source errors on the calculation results, the arithmetic mean of the two products was selected as the final result of GRACE satellite inversion and named as GRACE-mean solutions in the following discussions.

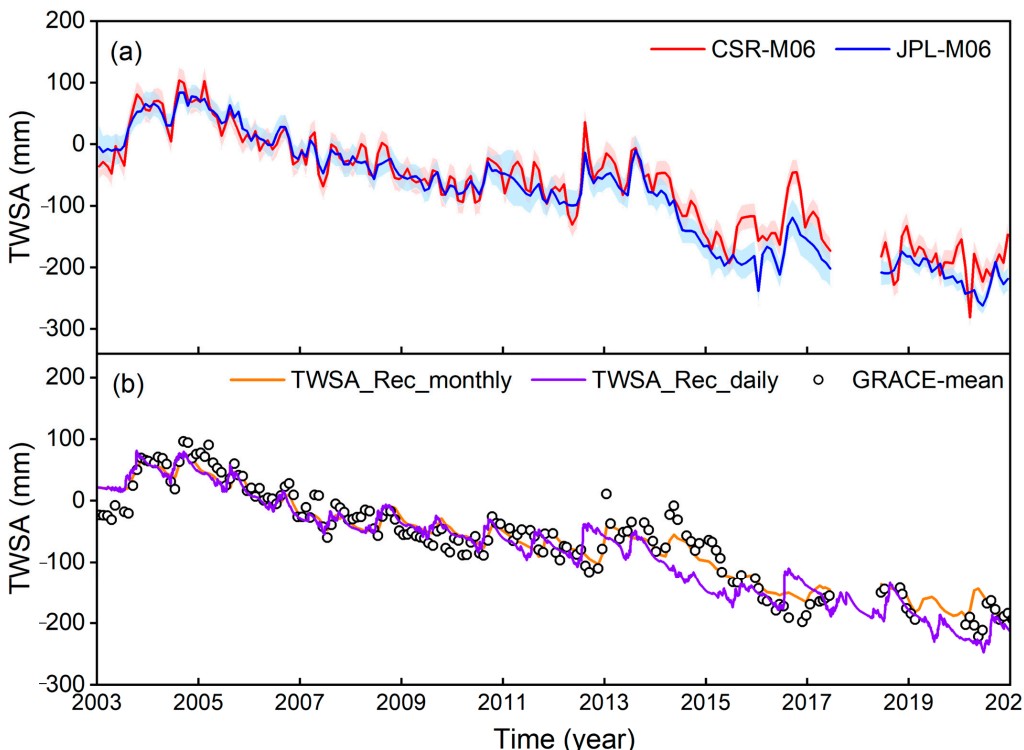

**Figure 5.** Comparison of different TWSA solutions: (**a**) time series of GRACE-derived TWSA during 2003–2020; (**b**) time series of reconstructed daily and monthly TWSA estimates over Hebei Province.

Figure 5b indicates the comparisons between the reconstructed and GRACE-derived solutions in Hebei Province. The climate-driven TWSA estimates were generated by combining CN05.1 products and reconstruction model at the daily scale. The monthly solutions were averaged based on the daily results. It can be seen that better consistency is observed between the reconstructed TWSA and GRACE solutions, with a correlation of up to 0.97. Thus, the reconstructed TWSA series provide reliable results to compensate for the missing months of GRACE satellites. We noticed that larger differences emerged for the former months of the study period, while this had little influence on the reconstructed TWSA results for the whole period.

*3.3. Temporal Characteristics of TWSA in Hebei Province*

In order to validate the properties of the reconstructed results, the BF-based results were also employed to quantify the discrepancies among different results. As shown in Figure 6a, it can be seen that TWSA estimates computed by this study show better consistency with GRACE solutions compared to the BF-based results. Specifically, the evaluation metrics between BF and GRACE solutions are 0.970, 0.940, and 21.15 mm for CC, NSE, and RMSE, respectively. Correspondingly, the performance was somewhat improved with the metrics of 0.971. 0.942, and 20.96 mm, respectively. Therefore, the data gaps between 2017–2018 were filled using the reconstructed results by this study so as to obtain the full-time TWSA series from 2003 to 2020.

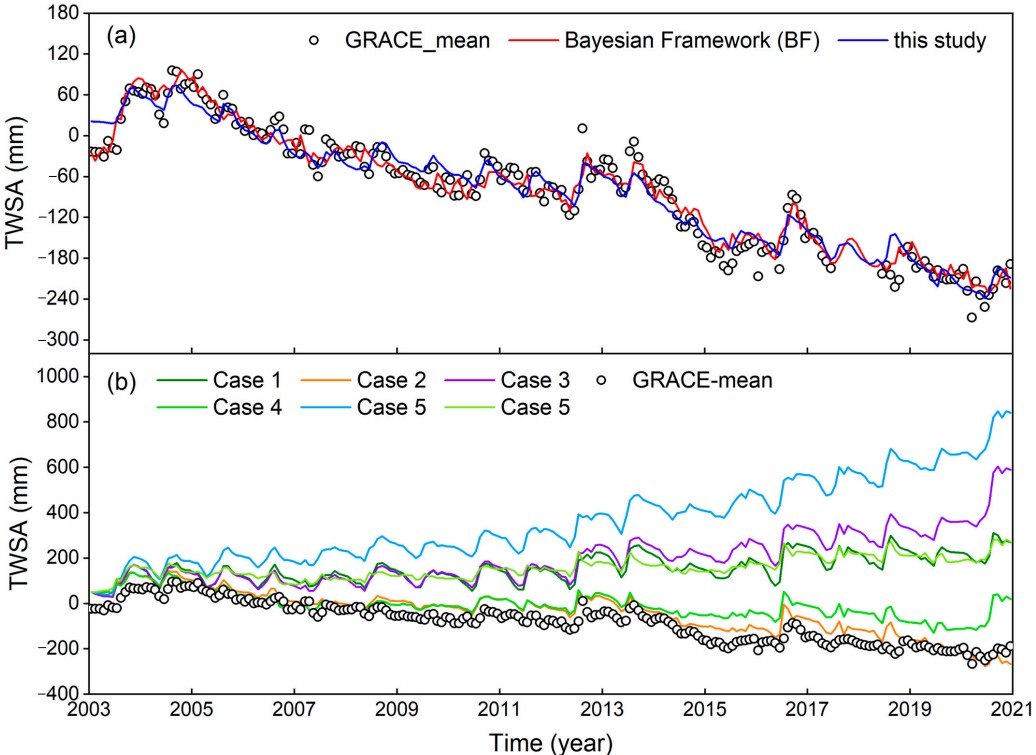

**Figure 6.** TWSA series in Hebei Province based on GRACE solutions and WB-based estimates from 2003–2020.

Due to the few differences between different meteorological products, six different combination scenarios are provided in Table 2. The TWSA series in Hebei Province were calculated under different combinations using the water-balance method, and the results are depicted in Figure 6b. The WB-based results under four combinations show a sharp upward trend, which are S1 (7.93 ± 0.56 mm/year), S3 (18.34 ± 0.78 mm/year), S5 (33.18 ± 0.74 mm/year), and S6 (6.86 ± 0.39 mm/year), respectively. The downward trend of the S2 case is close to that of the GRACE data, with the slope of −15.66 ± 0.45 mm/year and −16.75 ± 0.50 mm/year, respectively. The CC index between these two results is 0.95, indicating that the combination of CMA-Precp, Noahv2.1-ET, and Noahv2.1-runoff data can accurately capture the temporal characteristics of TWSA in the study area. The main reasons for the differences between the GRACE-derived and WB-based estimates are as follows: (1) GRACE-derived TWSA reflect the influence of human activities and climate factors, while the WB method only considers the climate factors; (2) it is challenging to acquire sufficient measured data in the actual study due to the large area and limited budget, and remote sensing products with some uncertainties are used in this study.

**Table 2.** List of different data combination scenarios based on the WB method.

| Scenarios | Precipitation | ET | Runoff | Slope (mm/year) |
| --- | --- | --- | --- | --- |
| Case 1 | CMA | GLEAM | Noahv2.1 | 7.93 ± 0.56 * |
| Case 2 | CMA | Noahv2.1 | Noahv2.1 | −15.66 ± 0.45 * |
| Case 3 | GPM | GLEAM | Noahv2.1 | 18.34 ± 0.78 * |
| Case 4 | GPM | Noahv2.1 | Noahv2.1 | −7.94 ± 0.50 * |
| Case 5 | Noahv2.1 | GLEAM | Noahv2.1 | 33.18 ± 0.74 * |
| Case 6 | Noahv2.1 | Noahv2.1 | Noahv2.1 | 6.86 ± 0.39 * |

Note(s): * Indicates significant at the 0.01 level.

### 3.4. Spatial Distribution Characteristics of TWSA in Hebei Province

The period of December 2020 was chosen as the typical time point, and the spatial characteristics of TWSA in Hebei Province were characterized based on GRACE solutions and WB-based estimates, as shown in Figure 7. According to the GRACE-derived results (Figure 7a), the TWSA values in the whole province were negative, indicating that the water storages were decreasing for the whole study area. The loss in the southern region was more severe, with two main funnel-shaped drops; one emerged at the frontier between Hebei and Henan Provinces, and the other appeared near Cangzhou City. The WB-based TWSA estimates (Figure 7b) differ greatly from the GRACE solutions in spatial distribution and amplitude of change, especially in the northern mountainous areas. The reason is that Hebei Province is crowded by a higher terrain in the northwestern region and a lower terrain in the southeastern. The northern mountainous areas are mostly covered with rocks with poor permeability. Therefore, surface runoff is much stronger, and the water recharged by precipitation infiltration is relatively smaller. In addition, the distribution of the WB-based TWSA estimates is messy. Even in the high mountainous areas without the influence of human activities, the irregular distribution characteristics do not match the actual situation.

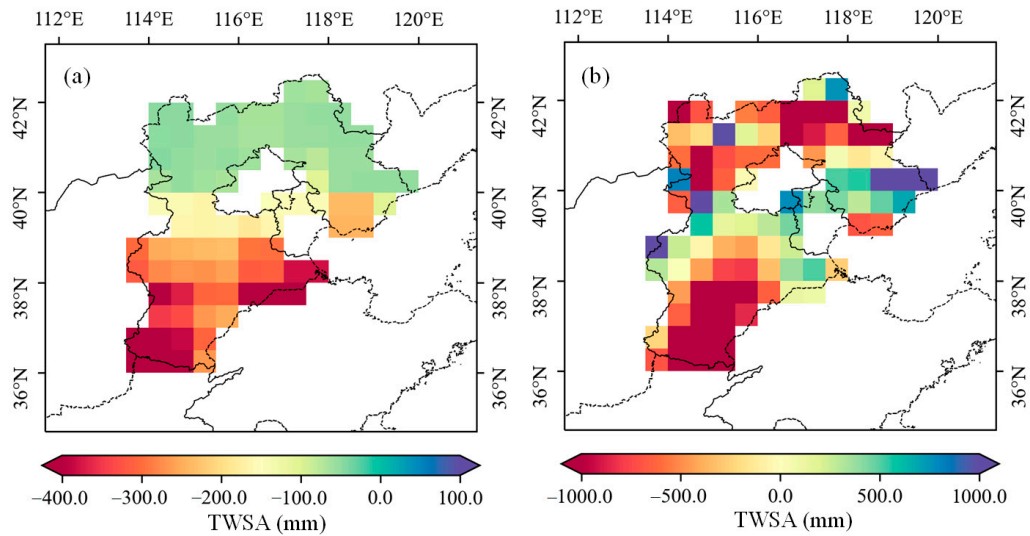

**Figure 7.** Spatial distribution of TWSA in Hebei Province in December 2020: (**a**) GRACE-based TWSA solutions; (**b**) WB-based TWSA estimates.

## 4. Discussion

### 4.1. The Influence of the SNWD Project on Plain and Mountain Regions

Based on the above two methods, the changes in TWSA in plain regions, mountain regions, and Hebei Province were determined from 2003 to 2020. The results are depicted in Figure 8, with yellow shades indicating the period after the operation of the central route of

the SNWD project (2015–2020, marked with yellow band). The results calculated by these two methods show large differences in plain and mountain regions. Both methods show that the downward trend slowed down after the arrival of the water from the SNWD project. As for the plain regions, the downward trend obtained by GRACE data is approximately twice that obtained by the WB-based estimates. The main reason is that human activities are intense in the plain regions, but the current WB-based estimates do not consider the influence of human activities, resulting in obvious underestimation problems. Table 3 lists the slopes of TWSA change before and after the SNWD project as calculated by the two methods. According to the GRACE inversion results, the slope in the plain areas decreased from $-15.65 \pm 0.99$ mm/year to $-15.13 \pm 2.31$ mm/year after water diversion, indicating that the downward trend was slightly alleviated. However, the WB-based TWSA estimates are the opposite, with the slope increasing from $-1.78 \pm 1.16$ mm/year to $-38.34 \pm 3.35$ mm/year.

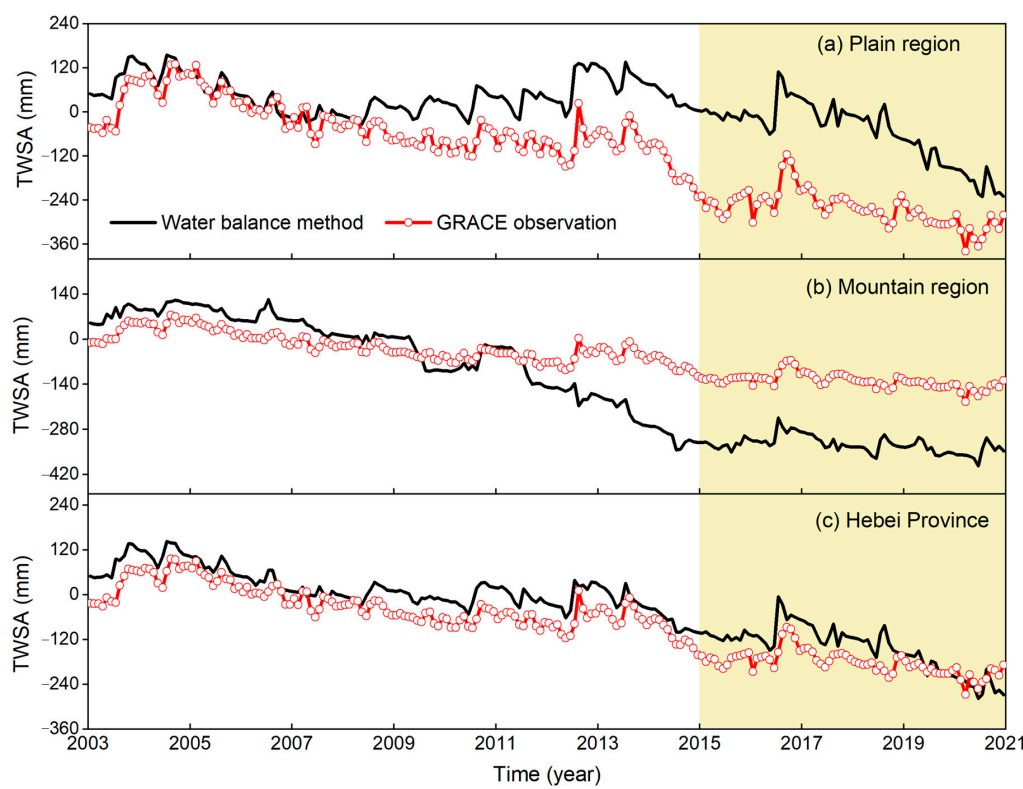

**Figure 8.** Curves of TWSA in different areas from 2003 to 2020: (**a**) plain region, (**b**) mountain region, (**c**) Hebei Province.

**Table 3.** Slope of TWSA before and after water diversion from the SNWD project by two methods.

| Area | Method | CC | 2003–2014 | 2015–2020 | 2003–2020 |
|---|---|---|---|---|---|
| Plain region | WB | 0.75 | $-1.78 \pm 1.16$ ** | $-38.34 \pm 3.35$ * | $-9.53 \pm 0.79$ * |
| | GRACE | | $-15.65 \pm 0.99$ * | $-15.13 \pm 2.31$ * | $-19.85 \pm 0.57$ * |
| Mountain region | WB | 0.91 | $-34.77 \pm 1.10$ * | $-7.03 \pm 1.57$ * | $-31.66 \pm 0.62$ * |
| | GRACE | | $-6.94 \pm 0.45$ * | $-3.72 \pm 1.45$ * | $-6.97 \pm 0.25$ * |
| Hebei Province | WB | 0.95 | $-12.53 \pm 0.77$ * | $-28.14 \pm 2.54$ * | $-15.66 \pm 0.45$ * |
| | GRACE | | $-12.81 \pm 0.79$ * | $-11.41 \pm 1.86$ * | $-16.75 \pm 0.50$ * |

Note(s): * Indicates significant at the 0.01 level; ** indicates significant at the 0.05 level.

With respect to the Hebei Province, the curves of TWSA calculated by the WB-based estimates and GRACE solutions show similar seasonal and long-term characteristics, with the CC of 0.95 and slopes of $-16.75 \pm 0.50$ mm/year and $-15.66 \pm 0.45$ mm/year, respectively. Both results demonstrate that TWSA estimates in Hebei Province exhibit an overall and pronounced declining pattern from 2003 to 2020. Considering the influence of the SNWD project, the slope of TWSA by the GRACE solutions has decreased in Hebei Province, slowing the downward trend by 10.93%, especially in the mountains by 46.39%, which indicates that the loss of TWSA in the area has been alleviated to some extent. The downward trends of the WB-based estimates increased by about twice, indicating that the TWS continued to lose after water diversion, which is mainly caused by the serious overestimation of the TWSA trend by the method in the plain regions.

Comparing the results of the two methods, it can be concluded that the results based on GRACE solutions are more consistent with the actual situation compared to the WB-based estimates. After the implementation of the SNWD project, the loss of TWSA in Hebei Province was alleviated to some extent, and the TWSA is in a relatively stable state. Meanwhile, as of the end of 2020, the trend of water resource depletion has not yet been reversed. It is worth noting that some pixels have only approximately half of their areas within the study area due to the geometric shape of the study area and the coarse spatial resolution of GRACE data. Therefore, whether these pixels are considered or not, it will inevitably cause some uncertainties when calculating the changes in water storage for the entire Hebei Province.

*4.2. The Influence of the SNWD Project on the Spatial Distribution Characteristics of TWSA in Hebei Province*

For investigating the influence of the SNWD Project on TWSA trends within Hebei Province, the long-term trends were calculated using GRACE solutions and the WB-based estimates before (2003–2014) and after (2015–2020) water diversion, respectively. Figure 9 illustrates the spatial distribution of TWSA trends in Hebei Province during different periods. It can be observed that GRACE solutions show that the declining trends of TWSA progressively increased from northern to southern regions, with the slope changing from $-14.26 \pm 0.42$ mm/year to $-28.16 \pm 0.56$ mm/year. After the water diversion, the downward trends were reversed, with a significant increasing trend in the northern regions. As can be seen from Figure 9c,d, the spatial distribution of WB-based TWSA estimates is rather chaotic. After the water diversion, the trend in the northern mountainous areas coincided with the GRACE solutions, while the opposite occurred in the central and southern plain areas, with more significant magnitude of long-term trends.

Comparing the spatial distribution of TWSA trends, it can be concluded that GRACE-derived TWSA are more accurate relative to WB-based estimates. The loss of TWSA gradually increases from northern to southern regions in the province. After the SNWD project, the water-shortage problem in Hebei Province was alleviated to some extent, especially in the northern regions, but the effect on the central and southern plain areas was not significant. The reason is that the plain area is characterized by a large number of agricultural zones, and large amounts of water resources are exploited for agricultural irrigation. Therefore, the water resources in these regions are seriously scarce, with large groundwater drop funnels. Water diversion cannot reverse the trend of large-scale groundwater depletion in a short period of time. Correspondingly, the northern mountainous regions are less impacted by human activities, and the improvement of water resources was relatively significant [38].

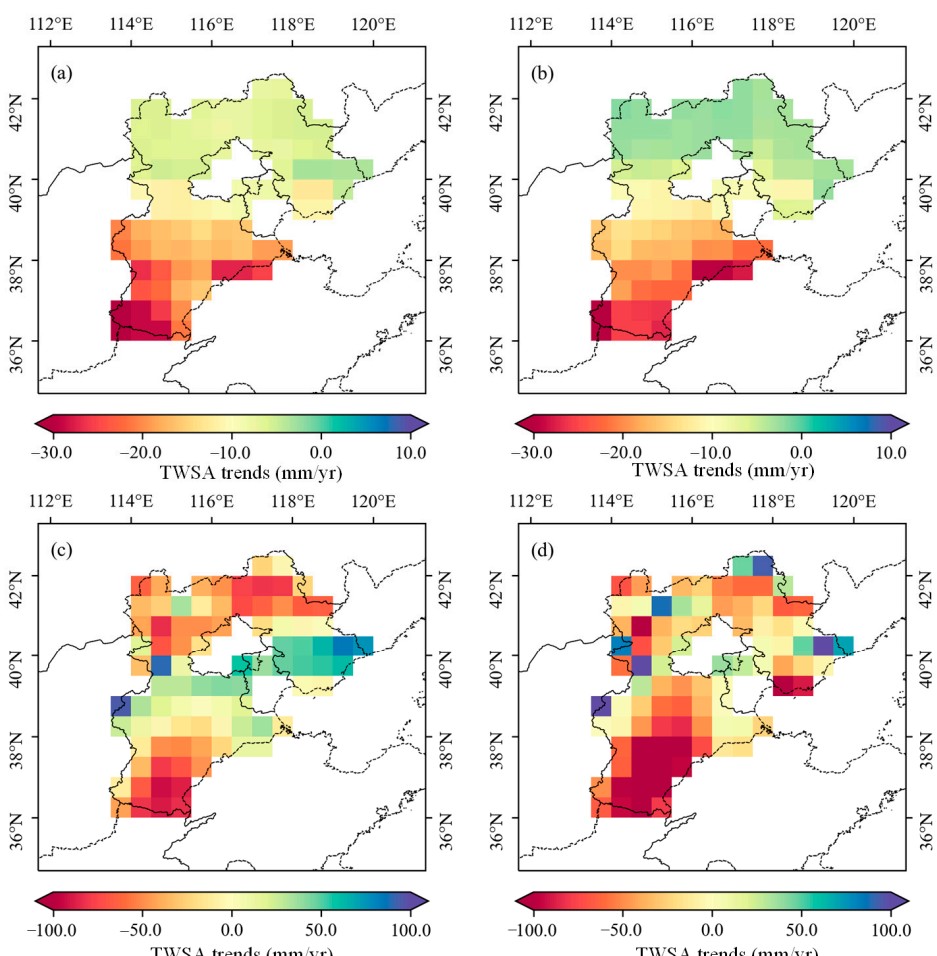

**Figure 9.** Spatial distribution of TWSA trends in Hebei Province: (**a**) GRACE solutions from 2003 to 2014, (**b**) GRACE solutions from 2015 to 2020, (**c**) WB-based estimates from 2003 to 2014, and (**d**) WB-based estimates from 2015 to 2020.

## 5. Conclusions

Accurate estimation of TWSA is significantly prominent for water management. In this study, the spatiotemporal characteristics of TWSA were estimated from 2003 to 2020 in Hebei Province using GRACE solutions and the WB-based estimates, respectively. The discontinuous TWSA series (2017–2018) between GRACE and GRACE-FO missions were filled by employing climate-driven models and CN05.1 meteorological products. Moreover, the influence of the SNWD project on the spatiotemporal TWSA was quantitatively analyzed. By analyzing the applicability and accuracy of the two methods, this study provides a foundation for the scientific management and rational planning of water resources in Hebei Province. The main conclusions are as follows:

The reconstructed TWSA estimates based on climate-driven models match well with GRACE solutions, with the CC, NSE, and RMSE of 0.971. 0.942, and 20.96 mm, which perform better than the BF-based results. Thus, it can provide comparable estimates to fill the data gaps between two missions. The accuracy of the WB-based TWSA was generated and evaluated under different data source combinations, which shows that the combination of CMA precipitation data and Noah v2.1 ET and runoff data has the highest consistency with the TWSA inverted by GRACE solutions, with a correlation coefficient of 0.95 and a slope error within 7%.

In terms of temporal evolution, the GRACE-derived and WB-based TWSA series are basically consistent, with a CC up to 0.95. From 2003 to 2020, the TWSA in Hebei Province showed an overall loss year by year, especially in the plain regions, which continue to be in serious loss. Spatially, the results of the two methods differ greatly. The GRACE

solutions show that the TWSA trends in Hebei Province increase from northern to southern regions, from −14.26 ± 0.42 mm/year to −28.16 ± 0.56 mm/year, indicating that the loss is gradually intensifying.

After the operation of the SNWD central route project, the GRACE solutions show that the TWS loss in Hebei Province was alleviated to some extent, but it was still in a state of loss by the end of 2020. Spatially, the downward trend of TWSA in the whole province was alleviated to varying degrees, especially in the mountain areas, which slowed down by 46.39%. The water-diversion project alleviated the shortage of water resources in Hebei Province to some extent but cannot reverse the trend of water-storage loss in the central and southern plain areas in a short period of time.

This research can provide some suggestions for future research and implications for decision makers. Firstly, the water-balance method currently only considers natural factors, which makes it unable to accurately assess the characteristics of water-storage changes. To enhance accuracy, it is necessary to incorporate anthropogenic factors such as the water usage and water diversion. Secondly, although the SNWD project alleviated the downtrends to some extent, it has not been able to reverse the overall downward trend. Therefore, the protection of water resources remains a long-term and formidable task.

**Author Contributions:** Conceptualization and methodology, T.L., D.Z., and Y.S.; writing—original draft preparation, T.L., Y.S., and J.S.; writing—review and editing, T.L. and D.Z.; supervision, X.Z., Y.S., J.S., and Y.L.; funding acquisition, X.Z., D.Z., and Y.S. All authors have read and agreed to the published version of the manuscript.

**Funding:** The research was supported in part by the National Key Research and Development Project (No. 2021YFC3000202), in part by the Key R&D Program of Hebei Province (21374201D), in part by the National Natural Science Foundation of China (42007176), and in part by the Natural Science Foundation of Shandong Province (ZR2020QD125).

**Institutional Review Board Statement:** Not applicable.

**Informed Consent Statement:** Not applicable.

**Data Availability Statement:** The JPL mascon data are available at https://grace.jpl.nasa.gov/data/get-data/jpl_global_mascons/#opennewwindow (accessed on 17 May 2022). The CSR mascon data are available at http://www.csr.utexas.edu/grace (accessed on 5 May 2022). The NOAH model is available at https://disc.gsfc.nasa.gov/datasets?keywords=GLDAS (accessed on 2 August 2022). No new data were created or analyzed in this study. Data sharing is not applicable to this article.

**Acknowledgments:** We are grateful to CSR and JPL for providing the monthly GRACE gravity field solutions; the Goddard Space Flight Center for providing the monthly GLDAS-2.1 data; the China National Meteorological Science Data Center for providing the monthly precipitation products; and the Global Land Evaporation Amsterdam Model for providing the ET data. The editor and reviewers are sincerely acknowledged for their instructive and detailed comments on this manuscript. The authors appreciate all centers for supplying the open-source datasets used in the present research.

**Conflicts of Interest:** The authors declare no conflict of interest.

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
