# Peer review of "The Influence of the South-to-North Water-Diversion Project on Terrestrial Water-Storage Changes in Hebei Province"

_water, doi:10.3390/w15173112_

Round 1
Reviewer 1 Report
This paper mainly focusses on the utilizes GRACE solutions and WB-based estimates to analyze the spatiotemporal characteristics of TWSA from January 2003 to December 2020, and quantitatively evaluate the influence of SNWD central route in alleviating the loss of TWS in Hebei province.
The quality of the research is qualified and significant for understanding. The paper is recommended for publication after fixing several improvements of minor revision. Notably, the paper needs to explain the problem it is trying to solve in the introduction. Comments follow.
Comments follow.
#1. 2.2.1 ,2.2.2 and 2.2.3, the authors choose the three types with 8 series data sets. But the different spatial resolution of the these data sets should be considered. Please clarify how these two data sets be used in fusion.
#2. 2.3.2, climate-driven TWSA estimates are introduced in this work, please clarify this operation and why in more detailed?
#3. In section 4.1 and line fig8. The result is interesting. The influence of the SNWD Project on plain and mountain regions are obviously different. If add analysis from scientific view, that will be much better.
#4. The discussion and conclusions should add several points in scientific view, which can help to condense your scientific contributions.
#5. There exit many texts expression in the full manuscript, and should be carefully refined and smoothed to be native English expression.
#6. Some figures in the manuscript needs to improve the quality according to the journal’s guide, especially the Figure 5 and 6.
There exit many texts expression in the full manuscript, and should be carefully refined and smoothed to be native English expression.
Author Response
Thank you very much for your efforts and time in reviewing our manuscript and your critical comments. We considered each comment raised from you carefully. We sincerely hope that our responses will make you satisfied.

Reviewer 2 Report
Major comments:
The authors present an interesting study that quantifies the impact of the South-to-North Water Diversion (SNWD) Project on alleviating the water storage deficit in Hebei province. They utilize reconstructed GRACE-based and WB-based TWSA estimates to analyze this influence. The problem of applying the reconstructed GRACE data to investigate TWSA over areas with intense human activity should be of interest to the readership of this journal. At this stage I recommend a Moderate Revision from Water. The following specific major or minor comments may be helpful for the authors in improving the manuscript.
1. In the Abstract Section, the authors made one of important conclusions that “After the SNWD project, GRACE-derived results show that the downtrends of TWSA are reduced by 10.93%, especially in mountainous areas by 79.78%”. However, the explanation provided in lines 420-422 is too simplistic. To enhance the persuasiveness of the conclusions, I suggest conducting a more comprehensive analysis of this phenomenon, considering aquifer characteristics and other relevant factors.
2. I think that the geometric shape of certain parts of the study area is too narrow (Figure 1). This could introduce significant uncertainties in GRACE derived TWSA, given the spatial resolution limitations of GRACE data (approximately 300 km). Therefore, I would suggest the authors to discuss the potential impact of these uncertainties on their conclusions.
3. The authors stated that “We notice that larger differences emerge on the former months of the study period, while it has little influence on the reconstructed TWSA results for the whole period (lines 302-304)”. To strengthen this conclusion, I recommend conducting a quantitative comparison between the climate-driven TWSA estimates computed in this study and the BF-based results (shown in Figure 6a), focusing on long-term trends before and after the SNWD project.
4. Please place the abbreviation "GRACE" in line 51 instead of lines 56-57.
5. Line 339: Delete “, ** indicates significant at the 0.05 level”.
6. Add units to Figure 7 and Figure 9 for clarity.
Author Response

(The authors gave the same response as above.)

Reviewer 3 Report
Dear All,
Please find attached my comments on the paper.
Best Regards.

Author Response

(The authors gave the same response as above.)
